# NTFields: Neural Time Fields for Physics-Informed Robot Motion Planning

**Ruiqi Ni**
Department of Computer Science
Purdue University
`ni117@purdue.edu`

**Ahmed H. Qureshi**
Department of Computer Science
Purdue University
`ahqureshi@purdue.edu`

## Abstract

Neural Motion Planners (NMPs) have emerged as a promising tool for solving robot navigation tasks in complex environments. However, these methods often require expert data for learning, which limits their application to scenarios where data generation is time-consuming. Recent developments have also led to physics-informed deep neural models capable of representing complex dynamical Partial Differential Equations (PDEs). Inspired by these developments, we propose Neural Time Fields (NTFields) for robot motion planning in cluttered scenarios. Our framework represents a wave propagation model generating continuous arrival time to find path solutions informed by a nonlinear first-order PDE called the Eikonal equation. We evaluate our method in various cluttered 3D environments, including the Gibson dataset, and demonstrate its ability to solve motion planning problems for 4-DOF and 6-DOF robot manipulators where the traditional grid-based Eikonal planners often face the curse of dimensionality. Furthermore, the results show that our method exhibits high success rates and significantly lower computational times than the state-of-the-art methods, including NMPs that require training data from classical planners. Our code is released: `https://github.com/ruiqini/NTFields`.

## 1 Introduction

Motion Planning (MP) is one of the core components of an autonomous robot system that aims to interact physically with its surrounding environments. MP algorithms find path solutions from the robot's start state to the goal state while respecting all constraints, such as collision avoidance. The quest for fast, scalable MP methods has led from traditional approaches such as RRT* (LaValle et al., 2001), Informed-RRT* (Gammell et al., 2014), and FMT* (Janson et al., 2015) to NMPs that exhibit promising performance in high-dimensional spaces. However, a significant bottleneck in state-of-the-art NMPs is their need for expert trajectories from traditional MP methods, limiting their application to high-dimensional scenarios where large-scale data generation is time-consuming.

Recent developments have provided us with ways to have physics-informed deep learning models (Raissi et al., 2019; Li et al., 2020; Smith et al., 2020) that can directly solve complex PDEs such as Navier-Stokes, Burgers, and Schrodinger equations. Inspired by these neural PDE solvers and to overcome the expert data needs of NMPs, this paper introduces NTFields for robot motion planning in cluttered environments. Our NTFields are generated by a physics-informed neural model driven by a first-order, non-linear PDE called the Eikonal equation, whose solution represents the shortest arrival time from a source to a destination location under a pre-defined speed model, and leads to the continuous shortest-path between two locations (Sethian, 1996; Clawson et al., 2014). Our model generates a continuous time field between the robot's given start and goal configurations while respecting collision-avoidance constraints, leading to a path solution in sub-seconds.

Our method's salient features and contributions are summarized as follows: 1) A novel physics-informed PDE solving the Eikonal equation formulation for robot motion planning under collision-avoidance constraints in high-dimensional spaces. 2) A novel neural architecture design that encapsulates various properties of our PDE, resulting in a scalable, fast NMP method. 3) A novel bidirectional algorithm that quickly finds path solutions by iteratively following the gradient of neural time fields, marching towards each other from the start and goal configurations. 4) Unlike prior,

state-of-the-art NMP methods requiring extensive expert motion trajectories, NTFields only require randomly sampled robot start and goal configurations and learn robot motion time fields by directly solving the PDE formulation to find path solutions in sub-second times. Our data generation takes less than 3 minutes, even in complex high-dimensional scenarios, and yet, our method computes paths significantly faster than traditional planners while retaining high success rates. 5) We demonstrate our method in various 3D environments, including the Gibson dataset, and also solve motion planning problems for 4-DOF and 6-DOF robot manipulators where traditional Eikonal equation based planners such as the Fast Marching Method (FMM) (Sethian, 1996) struggle due to computational time complexity. 6) Finally, we compare our method against state-of-the-art NMP methods, which require expert motion trajectories, and against existing Neural PDE solvers for the Eikonal equation to highlight their limitations in solving complex motion planning problems.

## 2 RELATED WORK

Perhaps the most relevant work is the FMM (Sethian, 1996; Valero-Gomez et al., 2013; Treister & Haber, 2016), which numerically solves the Eikonal equation to determine the time field for robot motion planning. However, numerical approaches require discretization of configuration spaces, thus failing to provide the continuous-time fields and suffering from the curse of dimensionality in terms of computational tractability. There also exist other traditional methods for solving MP problems without relying on the Eikonal equation. These methods range from sampling-based MP (Karaman & Frazzoli, 2011; Janson et al., 2015; Gammell et al., 2015) to gradient-based optimization techniques (Ratliff et al., 2009; Kalakrishnan et al., 2011). The former techniques relatively scale better than FMM but still exhibit large computation times in high-dimensional space. In contrast, the latter methods do not provide global solutions and often lead to local minima.

Recent developments have introduced NMPs, which learn from expert data via imitation and improve the performance of classical techniques in higher-dimensional problems. For instance, (Ichter et al., 2018; Kumar et al., 2019; Qureshi & Yip, 2018) generate informed samples in the regions containing the path solution to guide the underlying classical MP methods. (Qureshi et al., 2019; 2020) generate end-to-end path solutions and revert to classical planners to inherit their completeness guarantees. (Ichter & Pavone, 2019) perform MP in latent spaces to find path solutions but lacks interpretability. In general, NMPs, despite their computational gains, usually rely on classical MP methods to provide the training data, which hinders their application in scenarios where data generation is challenging. Recent approaches (Ortiz et al., 2022; Liu et al., 2022; Saulnier et al., 2020; Finean et al., 2021) use gradient information from Neural Distance Field (NDF) (Chibane et al., 2020; Xie et al., 2021) for collision avoidance resulting in real-time robot motion generation methods. In a similar vein, (Adamkiewicz et al., 2022) uses gradient information from Neural Radiance Field (NeRF) (Mildenhall et al., 2020) for visual navigation. Other relevant and recent methods to our NTFields are Implicit Environment Functions (IEF) (Li et al., 2022) and cost-to-go (c2g) function (Huh et al., 2021), which also generate time fields or c2g function, then compute gradients for robot motion planning. However, IEF and c2g are not physics-driven methods; instead, they are supervised learning approach that requires training data from FMM or PRM. Note that, unlike prior NMP methods, our approach does not require motion path trajectories for imitation learning and instead directly solves the PDE formulated by the Eikonal equation to generate a continuous-time field for motion planning.

Recent deep learning advances encapsulate various physics-based continuous functions solving PDEs without relying on explicit numerical derivatives. These methods exploit the neural network's back-propagation for gradient computation of output concerning related inputs. Similarly, EikoNet (Smith et al., 2020) solves the Eikonal equation for modeling the time field concerning seismology. However, EikoNet, as we show in our experiments, does not generalize to motion planning tasks requiring speed constraints in obstacle regions. Instead, we introduce novel formulation and architecture design enabling continuous time field modeling through solving the Eikonal equation for complex robot motion planning tasks.

## 3 NTFIELDS: PROPOSED FRAMEWORK

This section presents our physics-informed NTFields framework (Fig. 1) and discusses its formulation, architecture design, training details, and overall bidirectional planning execution.

## 3.1 Robot Motion Planning and Eikonal Equation Formulation

Let robot configuration-space (c-space) be denoted as $\mathcal{Q} \subset \mathbb{R}^d$ with dimensions $d \in \mathbb{N}$, and its surrounding environment, often known as workspace, be denoted as $\mathcal{X} \subset \mathbb{R}^m$, with dimensions $m \in \mathbb{N}$. The workspace usually contains obstacles, forming formidable $\mathcal{X}_{obs} \subset \mathcal{X}$ and feasible $\mathcal{X}_{free} = \mathcal{X} \backslash \mathcal{X}_{obs}$ workspace, and their corresponding obstacle $\mathcal{Q}_{obs} \subset \mathcal{Q}$ and obstacle-free $\mathcal{Q}_{free} = \mathcal{Q} \backslash \mathcal{Q}_{obs}$ configuration space. The standard robot motion planning objective is to find a path in an obstacle-free c-space $\mathcal{Q}_{free}$ connecting a given start $q_s \in Q_{free}$ and goal $q_g \in Q_{free}$ configurations. In addition, it is often desirable for these paths to fulfill optimality conditions, such as having the shortest Euclidean distance or minimum arrival time from a given start to goal configurations. In practice, planning using time fields is preferred as it allows constraints on robot speed near the obstacles, leading to safe navigation (Herbert et al., 2017). However, since modeling the time field in configuration space is computationally expensive under speed constraints, especially in higher-dimensional problems, most existing methods impose distance-based optimality and rely on advanced path tracking techniques for robot safety around obstacles.

The Eikonal equation is a first-order, nonlinear PDE approximating wave propagation. Its solution is the shortest arrival time from a source to a destination location under a pre-defined speed model, which corresponds to the continuous shortest path problem (Sethian, 1996; Clawson et al., 2014). Let $S(x)$ and $T(x, y)$ represent the speed model on point $x$ and the corresponding wave arrival time from the given source $x$ to the target $y$. Then, the Eikonal equation relates the speed and arrival time as $\frac{1}{S(y)} = \|\nabla_y T(x, y)\|$, where $S(y)$ is the speed at target point $y$, and $\nabla_y T(x, y)$ is a partial derivative of the arrival time model with respect to the target point $y$. However, the solutions to the Eikonal equation have singularity near the source point when the arrival time is almost zero. This limitation is often resolved by a factored formulation represented as $T(x, y) = \|x - y\| \times \hat{\tau}(x, y)$ (Fomel et al., 2009; Treister & Haber, 2016; Smith et al., 2020), where $\hat{\tau}$ represents a factorized time field from time field. However, such factored representation is not suitable for motion planning problems as the speed inside obstacles, i.e., formidable space $\mathcal{Q}_{obs}$, needs to be zero, making arrival time infinity. In addition, we also observe that the arrival time model needs to obey the following symmetry property for motion planning between given start and goal configurations $T(q_s, q_g)=T(q_g, q_s)$. This implies that the arrival time from start $q_s$ to destination $q_g$ and vice versa are equal under the assumption that only one optimal path solution can exist connecting the given configurations. The symmetry property also applies to partial derivatives of the time field as $\nabla_{q_s} T(q_s, q_g) = \nabla_{q_s} T(q_g, q_s)$. Therefore, we factorize the arrival time $T$ in the following form and introduce a new factorized time field $\tau(q_s, q_g)$ as:

$$T(q_s, q_g) = \frac{\|q_s - q_g\|}{\tau(q_s, q_g)} \tag{1}$$

In the above equation, the model time field $T(q_s, q_g) = 0$ when $q_s = q_g$. As we require speed in formidable obstacle space to be almost zero, we can explicitly make $\tau(q_s, q_g) \to 0$ for any arbitrary configurations in obstacle space, i.e., $\{q_s, q_g\} \in \mathcal{Q}_{obs}$. Therefore, $T(q_s, q_g) \in [0, \infty)$ and by our factorization in Equation 1, $\tau(q_s, q_g)$'s value range from 0 to 1. Furthermore, the expansion of the Eikonal equation using the chain rule, with $T(q_s, q_g)$ as defined in Equation 1, results in the following formulation (see Appendix A for full derivation).

$$S(q_g) = \frac{\tau^2(q_s, q_g)}{\sqrt{\tau^2(q_s, q_g) + \|q_s - q_g\|^2 \cdot \|\nabla_{q_g}\tau(q_s, q_g)\|^2 - 2\tau(q_s, q_g) \cdot (q_g - q_s) \cdot \nabla_{q_g}\tau(q_s, q_g)}} \tag{2}$$

## 3.2 Neural Time Fields

We model the physics governed by Equation (2) using a deep neural network. Our neural architecture outputs the factorized time field $\tau$ for the given robot configurations' representation concerning the given environment. We also introduce novel neural architecture to respect the symmetric property of time field w.r.t motion planning, as described above. In addition, to avoid the numerical issue of the time field being $\infty$ caused by zero speed in obstacle space, we introduce a speed model respecting collision constraints.

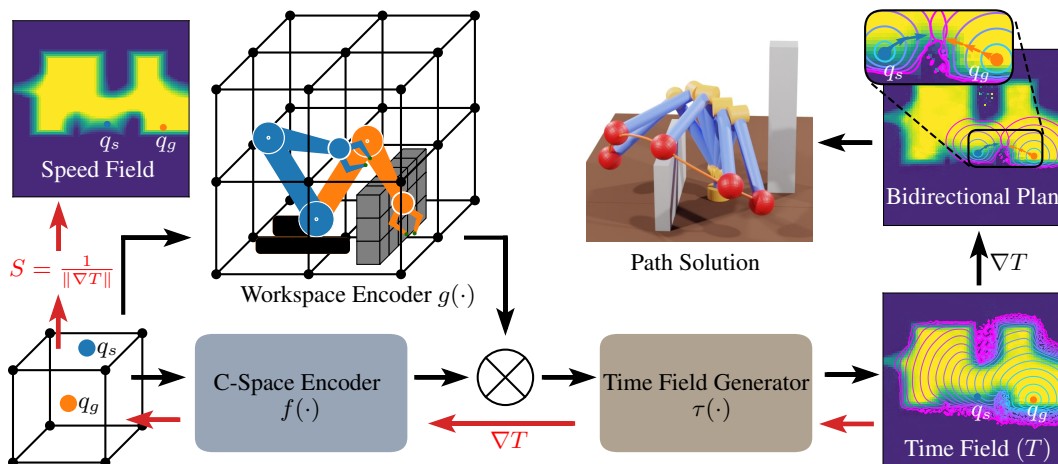

Figure 1: Our method consists of a C-Space encoder, workspace Encoder and time field generator. The C-space encoder encodes the start $(q_s)$ and goal $(q_g)$ configurations, whereas the workspace encoder provides environment-centric robot configuration representations. These c-space and workspace features are passed through the symmetric operator $\otimes$, forming the input for our time field generator. Finally, the gradient of the time field results in the speed field, and the final path is computed by following those gradients in a bidirectional manner. The black arrows show the forward propagation of our network, and the red arrows show the backpropagation to compute the loss.

**Speed Model $S^*(q)$:** We define our ground truth speed model, denoted as $S^*(q)$, at any robot configuration $q \in \mathcal{Q}$ as

$$S^*(q) = \frac{s_{const}}{d_{max}} \times \text{clip}(d_{\mathcal{M}}(\mathbf{p}(q), \mathcal{X}_{obs}), d_{min}, d_{max}), \tag{3}$$

where $\mathbf{p} \subset \mathbb{R}^m$ represents robot surface obtained via forward kinematics for the given configuration $q \in \mathcal{Q}_{free}$, and $d_{\mathcal{M}}$ computes a distance between robot surface $\mathbf{p}$ and obstacles $\mathcal{X}_{obs}$ in the workspace. The $d_{min}$ and $d_{max}$ are minimum and maximum distance thresholds, and $s_{const}$ is a user-defined speed constant. The clip functions bound the distance function with range $[d_{min}, d_{max}]$. We assume constant speed if a robot's distance from obstacles exceeds $d_{max}$.

**Robot configuration and workspace encoding:** For a given robot configuration $q \in \mathcal{Q}$, our encoders generate the robot-centric workspace embedding $g(q)$ and direct configuration embedding $f(q)$. The former enables reasoning about the relative position of obstacles and the robot, including collisions, for decision-making. Whereas the latter allows direct computation of the gradient of time for input configurations needed for solving the Eikonal equation.

*Workspace encoding $g(q)$:* We extend the procedure described in (Chibane et al., 2020) to obtain the robot-centric workspace embedding $g(q)$. First, we sample the sparse workspace obstacle point-cloud $\mathbf{X} \subset \mathcal{X}$ and convert it into a voxel grid $\mathbf{W}_0$ of dimension $N \times N \times N \times 1$, where the last dimension indicates each voxel's occupancy. Then, the $\mathbf{W}_0$ is passed through a 3D CNN layer to get the output $\mathbf{W}_1$ of size $N \times N \times N \times K$, where $K$ is the number of feature maps. Note that the input $\mathbf{W}_0$ and output $\mathbf{W}_1$ have the same first 3D dimensions, which we retain via padding. Next, we combine $\mathbf{W}_0$ and output $\mathbf{W}_1$ such that the resulting workspace feature representation $\mathbf{W}$ becomes of the size $N \times N \times N \times (K + 1)$. Note that our $\mathbf{W}$ is a multi-scale feature representation comprising both input and output layers of the 3D CNN module. To compute $g(q)$ based on $\mathbf{W}$ and $q$, we generate several samples $\{p_1, p_2, \cdots, p_n\} \in \mathbf{p}(q)$ on the robot surface $\mathbf{p} \subset \mathbb{R}^3$ using forward kinematics (FK) at the given configuration $q$. FK allows computing robot joint positions in the workspace using the robot joint angles. Therefore, given all joints' positions and robot geometry, i.e., links' lengths and widths, the robot workspace representation as a point cloud, $\mathbf{p}(q)$, is obtained. Note that the robot surface points are in the workspace, as is the obstacle point cloud. Therefore, using both robot surface point cloud and obstacle point cloud to obtain workspace encoding enables reasoning about the relative position of obstacles and the robot itself, including collisions, for decision-making. To do so, we compute the nearest grid cell in $\mathbf{W}$ for each point $p_i$. The grid cell is a 3D cube with

eight corners. Therefore the feature size for each grid cell is $8 \times (K + 1)$. Next, the feature vectors $8 \times (K + 1)$ are further combined via trilinear interpolation to form a representation for point $p_i$ of size $(K + 1)$. A full robot-centric workspace representation $g(q)$ becomes of size $n \times (K + 1)$.

*C-space encoding $f(q)$:* Finally, to compute a direct latent embedding of $q$, our c-space encoder uses a ResNet-style (He et al., 2016) feed-forward neural network, denoted as $f$. Since we need to compute the gradient of arrival time with respect to input configurations, the direct encoding of $q$ with MLP-based ResNet blocks allows gradient computation. In summary, the function $f$ takes the robot configuration $q$ as an input, passes them through ResNet blocks, and outputs latent embedding.

Hence, given a robot configuration, our encoders provide robot configuration embedding $f(q)$ and their robot-centric workspace representation $g(q)$, as shown in Fig. 1.

**Nonlinear symmetric operator:** To address the symmetry property of the Eikonal Equation, in this section, we introduce a nonlinear symmetric operator, denoted as $\bigotimes$, to combine the features $f(q_s)$, $f(q_g)$, $g(q_s)$, and $g(q_g)$ to form an input to our time fields generator predicting $\tau$. We choose a nonlinear symmetric operator min and max to combine the given feature vectors. However, since each of these operators loses some feature information, we observed that concatenating multiple operators' output leads to better performance. Therefore, we combine the robot configuration feature vectors in our setup by concatenating both min and max operations. Let the concatenation of two arbitrary vectors $\boldsymbol{a}$ and $\boldsymbol{b}$ be denoted as $[\boldsymbol{a}, \boldsymbol{b}]$. Then, our nonlinear operator $\bigotimes$ combines the arbitrary feature vectors $\boldsymbol{a}$ and $\boldsymbol{b}$ as $\boldsymbol{a} \bigotimes \boldsymbol{b} = [\max(\boldsymbol{a}, \boldsymbol{b}), \min(\boldsymbol{a}, \boldsymbol{b})]$. Therefore, our configuration feature vectors are combined as $[f(q_s) \bigotimes f(q_g), g(q_s) \bigotimes g(q_g)]$. In Appendix. B, we further discuss the details to motivate our choice of nonlinear symmetric operator.

**Time field generation:** Our time field generation function is a ResNet-style neural architecture that takes the robot's start and target configuration embeddings, i.e., $[f(q_s) \bigotimes f(q_g), g(q_s) \bigotimes g(q_g)]$, and outputs the factorized time field $\tau$. Our method learns to output $\tau$ and its partial derivative w.r.t inputs via backpropagation using our Eikonal-based physics model, i.e., Equation 2. The training procedure and the execution of our NTFields framework are described in the following.

**Training details:** We randomly sample $N \in \mathbb{N}$ different start and goal configuration pairs in the given environment and compute their speed field using our model $S^*(q)$ described in Equation 3. The resulting dataset to train NTFields is of form $\{(q_s, q_g, S^*(q_s), S^*(q_g))^1, (q_s, q_g, S^*(q_s), S^*(q_g))^2, \cdots, (q_s, q_g, S^*(q_s), S^*(q_g))^N\}$. The loss function to train NTFields is computed as follows. For a given start $(q_s)$ and goal $(q_g)$, NTFields outputs $\tau(q_s, q_g)$. Next, we use the predicted $\tau(q_s, q_g)$ to compute the corresponding speed values $S(q_s)$ and $S(q_g)$ using Equation 2. Finally, the NTFields' training loss between the ground truth speed values $[S^*(q_s), S^*(q_g)]$ and the predicted speed values $[S(q_s), S(q_g)]$ is defined as:

$$|1 - \sqrt{S^*(q_s)/S(q_s)}| + |1 - \sqrt{S^*(q_g)/S(q_g)}| + |1 - \sqrt{S(q_s)/S^*(q_s)}| + |1 - \sqrt{S(q_g)/S^*(q_g)}| \quad (4)$$

Our loss function is isotropic to prevent predicted speed from non-uniform errors due to different training sample pairs. Furthermore, we use a square root to smooth the loss function's gradient. We train our models end-to-end with their objective functions using the AdamW (Loshchilov & Hutter, 2017) optimizer. Additional training data generation details are available in Appendix Section D.

**Execution of bidirectional motion planning:** Once trained, we leverage our NTFields model to parametrize Equation 1, i.e., $T(q_s, q_g) = \|q_s - q_g\|/\tau(q_s, q_g)$, to compute the time field $T(q_s, q_g)$ and its partial derivatives $\nabla_{q_s} T(q_s, q_g)$ and $\nabla_{q_g} T(q_s, q_g)$. The norms of these partial derivatives relate to the speed model as $\|\nabla_{q_s} T(q_s, q_g)\| = 1/S(q_s)$ and $\|\nabla_{q_g} T(q_s, q_g)\| = 1/S(q_g)$, as described in Section 1. Since the speed model governs the magnitude of the gradient, the gradient step will be high when speed is low, especially near obstacles, leading to unsafe robot navigation. To mitigate such unsafe maneuvers, we multiple gradients with $S^2(q)$, leading to $\|S^2(q_s)\nabla_{q_s} T(q_s, q_g)\| = S(q_s)$ and $\|S^2(q_g)\nabla_{q_g} T(q_s, q_g)\| = S(q_g)$. Since our speed model is small near obstacles, the term $S^2(q)$ dynamically scales down the step size near obstacles. Furthermore, our model's symmetry behavior allows us to perform gradient steps bidirectionally from start to goal and from goal to start. Hence, we compute the final path solution bidirectionally using iterative gradient descent by updating the start and target configurations as follows, where $\alpha \in \mathbb{R}$ is a step size hyperparameter.

$$q_s \leftarrow q_s + \alpha S^2(q_s)\nabla_{q_s} T(q_s, q_g); \quad q_g \leftarrow q_g + \alpha S^2(q_g)\nabla_{q_g} T(q_s, q_g) \quad (5)$$

## 4 EXPERIMENTS

This section evaluates our approach through the following experiment analysis in solving a wide range of motion planning tasks.

(1) We analyze our architecture design by comparing our method with EikoNet (Smith et al., 2020) and the traditional FMM on a simple task. We also evaluate our choice of non-linear symmetry operator for feature embedding.

(2) We perform a comparative study and compare our approach with the state-of-the-art MP methods. **Baselines:** Our baselines comprise NMP and classical MP methods, including MPNet (Qureshi et al., 2019; 2020), IEF (Li et al., 2022), FMM (Sethian, 1996), RRT* (Karaman & Frazzoli, 2011) and lazy-PRM* (Bohlin & Kavraki, 2000). Our IEF3D stems from the IEF paper, both of which learn via supervising learning using the training data from FMM. However, IEF3D introduces the following modifications and upgrades to IEF: First, unlike IEF, which

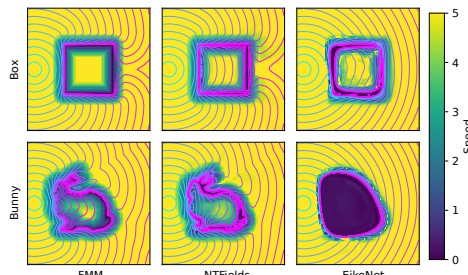

Figure 2: We compare our approach against FMM and EikoNet on box and bunny environment for time field generation. Our method is able to generate correct time fields for both scenes, whereas EikoNet fails under complex environment geometry.

used the grid points as goal configurations, IEF3D uses FMM's nearest numerical solutions on a fine-grained grid for randomly sampled start and goal pairs. Second, IEF3D uses 3DCNN for 3D environment encoding. Third, IEF3D uses update steps, as in Equation 5, to prevent large gradients leading to tangling paths. In contrast, IEF uses a multi-directional gradient search to select the direction with a lower distance to the goal, which prevents tangling paths but consumes larger computational times than IEF3D. **Evaluation Metrics:** Our evaluation metrics include planning time, path lengths, safety margin, and success rate. The planning time indicates the time it took for a planner to find a path solution, whereas the path length measures the sum of Euclidean distance between path waypoints. The safety margin indicates the closest distance to obstacles along the trajectory. Finally, the success rates represent the percentage of collision-free paths connecting the given start and goal found by a given planner.

(3) We conduct a scalability analysis of our method to high-dimensional manipulators' configuration spaces where methods like FMM and IEF3D are computationally expensive to execute and train, respectively.

### 4.1 ARCHITECTURE DESIGN ANALYSIS

This section analyzes our formulation and architecture design enabling physics-driven neural motion planning under collision-avoidance constraints. We compare the generated time field by NTFields (ours), EikoNet (Smith et al., 2020), and FMM, where FMM represents the optimal time field for benchmarking. The EikoNet, similar to ours, also parametrizes the Eikonal equation. However, unlike our framework (see Equation 1), EikoNet overcomes singularity using the standard factored formulation, i.e., $T(q_s, q_g) = \|q_s - q_g\| \times \hat{\tau}(q_s, q_g)$ (Smith et al., 2020), and does not consider c-space and workspace symmetric embeddings and speed ranges. Therefore, EikoNet represents an ablation of our framework to highlight the importance of our formulation and architectural designs.

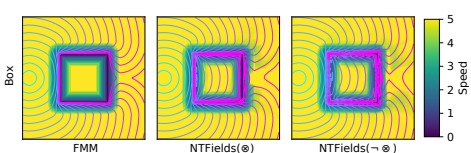

Figure 3: We compare NTFields with and without nonlinear symmetric operator against FMM on box environment for time field generation. NTFields with nonlinear symmetric operator recover better quality field.

We consider two 3D environments, named Box and Bunny, to represent simple and complex geometries for time field modeling. Fig. 2 shows all methods' speed and time field results, depicted for a single source point. The speed field is shown using the color intensity scale, where yellow and blue colors indicate the maximum and minimum speed, respectively. The contours represent the time field. From Fig. 2 contours, we see that NTFields perform similarly to FMM, successfully approximate

gradients for pathfinding, and ensure low speed near obstacle space. For EikoNet, they can predict the speed field in the box environment, but their time field appears as concentric circles in both scenarios. In summary, Fig. 2 highlights that the formulation presented in Section 3.1 for NTFields enables solving motion planning problems under-speed constraints. Furthermore, we also observe that our architecture can represent complex geometries, which are necessary for generalization to various start and goal configurations for generalizable MP.

We further analyze our NTFields framework's structure by evaluating the role of non-linear symmetry operators in recovering time fields. Fig. 3 shows the fields generated by FMM and NTFields with and without our non-linear symmetric operator. In NTFields, without a non-linear symmetry operator, denoted as $\mathrm{NTFields}(\neg \otimes)$, we concatenate all feature vectors, i.e., $[f(q_s), f(q_g), g(q_s), g(q_g)]$. It can be seen that our proposed operator $\mathrm{NTFields}(\otimes)$ recovers a better quality field, closer to expert FMM, whereas excluding it leads to distortion. Furthermore, the loss of $\mathrm{NTFields}(\otimes)$ and $\mathrm{NTFields}(\neg \otimes)$ at the same epoch were 2.3e-2 and 2.8e-2, respectively. Hence, we can conclude that our proposed non-linear symmetric operator improves performance in recovering time fields from source to destination.

## 4.2 COMPARATIVE ANALYSIS

This section presents a comparative analysis of our method and other baselines on cluttered 3D and Gibson environments.

**Cluttered 3D environment:** Our cluttered 3D environment (Fig. 4) comprises randomly placed ten obstacles of variable sizes and 2000 unseen start and goal pairs for solving motion planning tasks. We compare NTFields' performance with IEF3D, FMM, MPNet, RRT*, and LazyPRM*. The training data generation for each C3D environment took 3.74 seconds for NTFields, about 2.6 hours for IEF3D using FMM, and about 5 hours for MPNet (Qureshi et al., 2020) using RRT* to get comparable performances. For FMM, since it requires discretization, we select the nearest grid cells to our start and goal pairs for finding solutions. In the case of MPNet, we follow their standard training procedure using demonstration data from RRT* algorithm. Furthermore, we run RRT* and LazyPRM* on our test set until they find a path solution of length similar to MPNet's solution in the given time limit of 5 seconds. Fig. 4 shows the paths for a test start and goal pair where NTFields, IEF3D, FMM, MPNet, RRT*, and LazyPRM* path solutions are illustrated in orange, purple, green, blue, cyan, and yellow colors, respectively. It can be seen that the paths by NTFields (ours), IEF3D, and FMM are more smooth with sufficient safety margins compared to other methods. Additionally, the path lengths of our method are comparable to FMM since the Eikonal model governs both. However, the MPNet, LazyPRM*, and RRT* have shorter paths as both do not consider speed constraints near obstacles and often have sharp turns with lower safety margins. Note that the lower safety margins of LazyPRM*, RRT*, and MPNet's paths are considered unsafe for robot navigation (Herbert et al., 2017). The table in Fig. 4 shows the statistical result of six methods on our test dataset. The computation speed of NTFields is almost 6 times faster than MPNet and FMM, and over 10 times faster than RRT* and LazyPRM* in these cluttered scenarios. Although NTFields' computational speed is similar to IEF3D, the latter requires a large amount of training data for supervised learning. Furthermore, the success rates of all methods are high and similar, i.e., NTFields (ours) 99.3%, IEF3D 96.9%, MPNet 98.5%, FMM 99.8%, LazyPRM* 100%, and RRT* 100%. Finally, Appendix E provides an additional comparison analysis across eight different cluttered 3D environments.

**Large Complex Environments (Gibson):**

This section compares our method with IEF3D, FMM, RRT*, and LazyPRM* in cluttered, large Gibson environments (Fig. 6) to highlight our approach's scalability to home-like complex environments. We excluded MPNet as it performed slower than our method and IEF3D in our earlier comparison in cluttered 3D scenarios. Our test set contains randomly sampled 500 unseen start and goal pairs. Fig. 5 presents the speed field of our method and FMM in two environments. Our evaluation metrics for all methods are presented in Fig. 6 along with paths' depiction on a test start and goal pair. In the case of RRT* and LazyPRM*, we set the time limit of 5 seconds. Finally, the time to generate complete data was 2.2 minutes for NTFields and 2.6 hours for IEF3D.

The results demonstrate that NTFields recover similar, near-optimal speed fields as FMM in complex Gibson environments Fig. 5, leading to smooth path solutions while not requiring supervised training data. The paths shown in Fig. 6 are between start and goal pairs selected on two distant corners of the

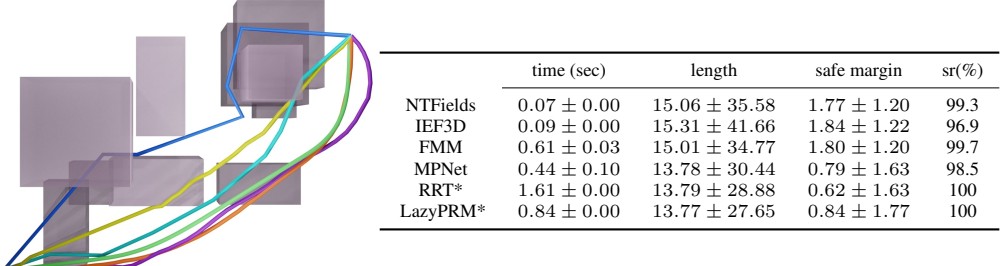

| | time (sec) | length | safe margin | sr(%) |
|---|---|---|---|---|
| NTFields | $0.07 \pm 0.00$ | $15.06 \pm 35.58$ | $1.77 \pm 1.20$ | 99.3 |
| IEF3D | $0.09 \pm 0.00$ | $15.31 \pm 41.66$ | $1.84 \pm 1.22$ | 96.9 |
| FMM | $0.61 \pm 0.03$ | $15.01 \pm 34.77$ | $1.80 \pm 1.20$ | 99.7 |
| MPNet | $0.44 \pm 0.10$ | $13.78 \pm 30.44$ | $0.79 \pm 1.63$ | 98.5 |
| RRT* | $1.61 \pm 0.00$ | $13.79 \pm 28.88$ | $0.62 \pm 1.63$ | 100 |
| LazyPRM* | $0.84 \pm 0.00$ | $13.77 \pm 27.65$ | $0.84 \pm 1.77$ | 100 |

Figure 4: Comparison in cluttered 3D environments. The left figure shows six paths generated by our method (orange), IEF3D (purple), FMM (green), MPNet (blue), RRT* (cyan), and LazyPRM* (yellow). The gray blocks are the obstacles. We make them slightly transparent for visualization. The right table shows statistical results on 2000 different starts and goals. Our method generates smooth paths with computational time about 6X faster than other methods besides IEF3D.

depicted environment. In this case, all methods except IEF3D find collision-free paths. According to our test set evaluation, NTFields and IEF3D perform similarly. The FMM outperforms all methods in success rate and safety margin, but computational times are almost 10 times slower than our method and IEF3D. The traditional techniques LazyPRM* and RRT* have the lowest safety margins and higher computational times. The success rate of RRT*, in this case, is the lowest as it takes significant computational time to perform collision checking for every path, resulting in failures concerning the time bound of 5 seconds. LazyPRM* performed faster with a higher success rate than RRT* because we used the precomputed graph in the former case. Overall, it can be seen that our method scales well to large complex environments, retains its low computational times, high safety margins, and success rates, and do not require supervised training data for learning.

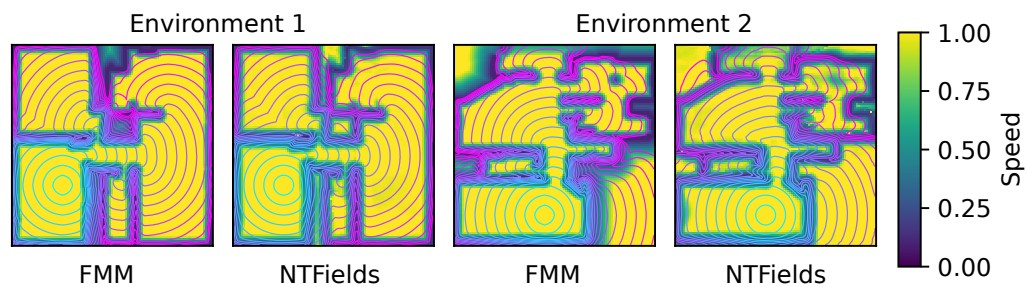

Figure 5: Speed Field and Time Field visualization, the colors show speed, and contour lines show arrival time from a source point, comparing NTFields (our) and FMM in two different, Gibson environments. Our method recovers expert-like fields required for path planning.

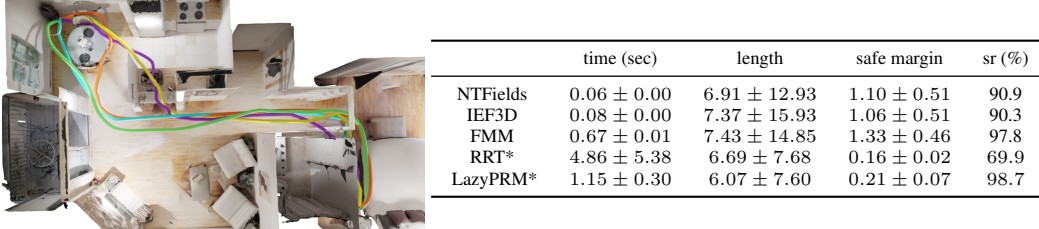

| | time (sec) | length | safe margin | sr (%) |
|---|---|---|---|---|
| NTFields | $0.06 \pm 0.00$ | $6.91 \pm 12.93$ | $1.10 \pm 0.51$ | 90.9 |
| IEF3D | $0.08 \pm 0.00$ | $7.37 \pm 15.93$ | $1.06 \pm 0.51$ | 90.3 |
| FMM | $0.67 \pm 0.01$ | $7.43 \pm 14.85$ | $1.33 \pm 0.46$ | 97.8 |
| RRT* | $4.86 \pm 5.38$ | $6.69 \pm 7.68$ | $0.16 \pm 0.02$ | 69.9 |
| LazyPRM* | $1.15 \pm 0.30$ | $6.07 \pm 7.60$ | $0.21 \pm 0.07$ | 98.7 |

Figure 6: (Left) Demonstration of paths computed by all methods in a large Gibson environment from a selected start and goal pair. The paths' colors are as follows: our method (orange), IEF3D (purple), FMM (green), RRT* (cyan), and LazyPRM* (yellow). (Right) Evaluation metrics depicting the performance of all methods on our test sets.

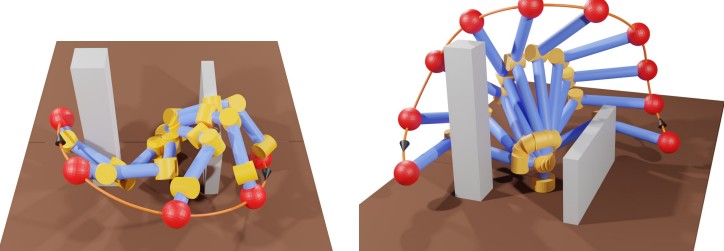

Figure 7: Demonstration of our method in solving motion planning problems of a 6-DOF (left figure) and 4-DOF (right figure) manipulator in 3D environments. The orange curve shows the final path with black arrows indicating robot transversal from start to goal configurations.

### 4.3 SCALABILITY ANALYSIS: HIGH-DIMENSIONAL ROBOT MANIPULATOR C-SPACE

This section demonstrates the NTFields' generalization to 4 and 6 DOF robot configuration spaces with 3D workspace obstacles (Fig. 7). The training data generation for these scenarios took around 60 seconds on our machine. The execution pipeline of this scenario is also highlighted in Fig. 1. We generate 100 random start and goal configuration pairs in each scenario for testing, where the training of our neural model is guided by Equation 2. We exclude other baselines due to their large computational times, especially FMM and IEF3D. The FMM is very computationally expensive to execute in high-dimensional spaces due to a very large number of grid cells. Consequently, the IEF3D is bottlenecked by its need for training data from FMM, which also limits its applicability to high-dimensional C-spaces (see Appendix E for more details). However, since our method directly solves the Eikonal equation without requiring the supervised training data from FMM, it generalizes and scales to higher-dimensional robot c-spaces. In the 4-DOF scenario (Fig. 7: Left), NTFields take about $0.08 \pm 0.00$ seconds with $96\%$ success rate; whereas in the 6-DOF scenario (Fig. 7: Right) it takes about $0.11 \pm 0.00$ seconds with $91\%$ success rate. Hence, our results show that our method successfully generates a continuous time field leading to path solutions in high-dimensional spaces. In contrast, it is computationally infeasible for grid-based methods like FMM and their data-driven methods like IEF3D or IEF to solve motion planning problems using the Eikonal equation.

## 5 CONCLUSIONS, LIMITATIONS, AND FUTURE WORK

We introduce a physics-informed neural motion planner that requires no expert demonstration data from classical planners and finds path solutions with significantly low computation times and high success rates in complex environments. Additionally, we formulate a physics-driven objective function and reflect it in our architecture design to directly parameterize the Eikonal equation and generate time fields for different scenarios, including a 6-DOF manipulator space, under collision-avoidance constraints. Finally, we compare our method with traditional algorithms and state-of-the-art learning-based motion planners (requiring expert data) and demonstrate up to 10X computational speed gains while retaining high success rates. Furthermore, our statistics also show that the data generation for our method takes at max 3 minutes in complex scenarios, whereas for other learning-based methods, it can take from hours to several days.

Our future work revolves around solving the limitations of our proposed work. First, currently, our method considers only the collision and speed constraints. Therefore, a possible avenue of research will be to explore other PDE formulations that have a similar property to the Eikonal equation and can encode more complex dynamical constraints such as acceleration. Second, our method only generalizes to the new start and goal configurations in a given environment. Hence, we also aim to extend our neural field method to generalize across different environments by exploring novel environment encoding architectures. Third, we also aim to validate our approach on real robot systems in our future work. Finally, since neural networks (NNs) are an approximation to PDE, there are few failure cases due to errors in learning the time-field model using the Eikonal equation. Therefore, another possible research direction could be to leverage a bag of neural networks to bootstrap the learning for improved performance.

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

APPENDIX

This appendix provides details on our model architectures, training and testing parameters, implementation of benchmark methods, and our computational device specifications.

## A  FACTORED EIKONAL EQUATION

Let $x$ and $y$ be the source and destination configurations. We have the Eikonal equation $\|\nabla_y T(x, y)\| = 1/S(y)$ and factored time $T(x, y) = \|x - y\|/\tau(x, y)$

$$\frac{1}{S(y)} = \|\nabla_y T(x, y)\| \tag{6}$$

$$\frac{1}{S(y)} = \|\nabla_y(\frac{\|x - y\|}{\tau(x, y)})\| \tag{7}$$

$$\frac{1}{S(y)} = \|\frac{\tau \nabla_y \|x - y\| - \|x - y\|\nabla_y \tau}{\tau^2}\| \tag{8}$$

$$\frac{1}{S(y)} = \|\frac{y - x}{\tau \|x - y\|} - \frac{\|x - y\|\nabla_y \tau}{\tau^2}\| \tag{9}$$

$$\frac{1}{S(y)} = \sqrt{\frac{1}{\tau^2} + \frac{\|x - y\|^2\|\nabla_y \tau\|^2}{\tau^4} - \frac{2(y - x) \cdot \nabla_y \tau}{\tau^3}} \tag{10}$$

$$S(y) = \frac{\tau^2}{\sqrt{\tau^2 + \|x - y\|^2\|\nabla_y \tau\|^2 - 2\tau(y - x) \cdot \nabla_y \tau}} \tag{11}$$

$$S(x) = \frac{\tau^2}{\sqrt{\tau^2 + \|x - y\|^2\|\nabla_x \tau\|^2 - 2\tau(x - y) \cdot \nabla_x \tau}} \tag{12}$$

## B  MODEL ARCHITECTURE

Our model has four parts: configuration space (c-space) encoder $f(\cdot)$, workspace (w-space) encoder $g(\cdot)$, nonlinear symmetric operator $\bigotimes$, and time field generator. The input to our framework includes the workspace point cloud, which we convert into a voxel grid $\mathbf{X}$, and the robots start $q_s$ and goal $q_g$ configurations in M-dimensional c-space. We use the forward kinematic function to compute the robot surface points associated with the given configuration. Furthermore, all environments' network architecture layer sizes are available in the supplementary source code. The network architecture flow and motivations are described as follows.

**C-space encoder:** It takes a robot configuration and passes them through fully connected (FC), non-linearity ELU, and several ResNet + ELU blocks (He et al., 2016) to generate the embedding $f(q)$.

**W-space Encoder:** To generate the configuration-dependent workspace encoding, we use the structure shown in Fig 1 (b). First, it uses 3D CNN (Conv3d) to extract two-level obstacle features $\mathbf{W}_0, \mathbf{W}_1$ from obstacle grid $\mathbf{X}$. Second, a trilinear interpolation function is used to extract the local, multi-scale features from $\mathbf{W}_0, \mathbf{W}_1$ for robot surface points $p(q)$. Finally, these local features are passed through an FC+ELU layer to compute the configuration-dependent workspace embedding $g(q)$.

**Nonlinear symmetric operator:** We highlight the two properties of the arrival time field and its gradient w.r.t source to motivate our choice of a nonlinear operator instead of a linear operator for combining features. First, assuming there is only one optimal path solution to a motion planning problem in a static environment, the time field between source $(q_s)$ and destination $(q_g)$ and its partial derivatives for the source needs to exhibit the symmetry property i.e., $\tau(q_s, q_g) = \tau(q_g, q_s)$. Second, the time field $\tau(q_s, q_g)$ and its gradient w.r.t source $q_s$ changes when $q_g$ changes. Therefore, to explicitly respect both properties, we introduce our nonlinear operator. For brevity, we introduce a

few extra notations. Let $u(q_s)$ and $u(q_g)$ be arbitrary robot start and goal feature vectors. Let $h$ be a neural network that takes the combined features $u(q_s) \bigotimes u(q_g)$ as an input and outputs the time field, i.e., $\tau(q_s, q_g) = h[u(q_s) \bigotimes u(q_g)]$. Then, their gradient w.r.t to start configuration $u(q_s)$ becomes $\nabla_{q_s} \tau(q_s, q_g) = \nabla_{u(q_s) \bigotimes u(q_g)} h \times \nabla_{q_s} [u(q_s) \bigotimes u(q_g)]$. Therefore, if operator $\bigotimes$ is linear (such as summation or average), the target term $u(q_g)$ will be independent of the start $u(q_g)$ and disappear in the gradient equation. Hence, if the target configuration changes, the $\nabla_{q_s} \tau(q_s, q_g)$ will not change. Thus linear operator violates the second property described earlier. In contrast, the nonlinear operator leads to $\nabla_{q_s} \tau(q_s, q_g)$ that depends on both source and target features, thus satisfying both symmetry and gradient properties of the arrival time field.

**Time field generator:** It takes the robot start and goal embedding $f(q_s), f(q_g), g(q_s), g(q_g)$ and passes them through the symmetric operator and several FC+ELU and ResNet+ELU layers to generate the time field $T$ (Fig 1 (c)). For symmetric operator, we concatenate $\max(f(q_s), f(q_g))$, $\min(f(q_s), f(q_g)), \max(g(q_s), g(q_g)), \min(g(q_s), g(q_g))$.

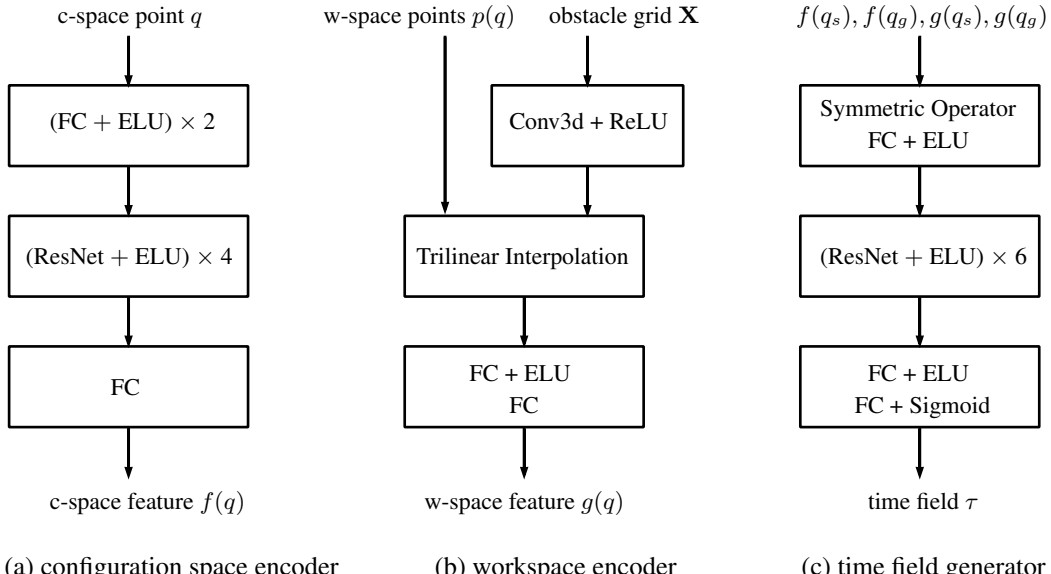

(a) configuration space encoder      (b) workspace encoder      (c) time field generator

Figure 8: NTFields neural architecture: FC is a fully connected layer, ResNet is a Residual Network block, and Conv3d is a 3D CNN layer. (a): $q$ is configuration space point, and $f(\cdot)$ is a c-space sample embedding; (b): $p(q)$ denotes robot surface points, $\mathbf{X}$ is voxel representation of obstacle point cloud, and $g(\cdot)$ is a configuration depended w-space encoding; (c): $\tau$ is factorized time field from time field.

## C   DATA PRE-PROCESSING

Our training data contains sampled start and goal pairs in robot c-space randomly. We first normalize robot c-space to $[-0.5, 0.5]$ on each dimension. Then we define the c-space sample range as a hypercube, i.e., all dimensions have the same range scale. Furthermore, we compute the speed for those configurations using our speed model. Our workspace point cloud includes 20000 points on the workspace obstacle surface, which we convert into $128 \times 128 \times 128$ occupied voxel grid following the procedure described in (Chibane et al., 2020). Finally, our pre-processing and training hyper-parameters are shown in our released code.

## D   TRAINING DETAILS

During training, we use weighted random sampling to form batches for our model training during each epoch. The traditional way of forming training batches is randomly sampling a small set of training data for each epoch. However, we observed that weighted random sampling of data with sample weights inversely proportional to Euclidean distance between start and goal pairs leads to

better performance than traditional random sampling. To train our NTFields framework end-to-end, we gradually introduce our workspace encoder as the training epochs increase using a scalar weight $\lambda$, i.e., $[f(q_s) \bigotimes f(q_g), \lambda g(q_s) \bigotimes \lambda g(q_g)]$. The $\lambda$ is set to 0 for the first $l_0$ epochs, then increased linearly to 1 between epoch $l_0$ to $l_1$, and then remains at 1 for the rest of the epochs. We observed that gradually increasing $\lambda$ stabilizes the training process, preventing it from converging to local minima. We choose $l_0 = 500, l_1 = 1000$ for Gibson scenes.

## E    Experiment Details

For benchmark methods, we use OMPL's RRT*, LazyPRM* implementation (Şucan et al., 2012), MPNet's open source code (Qureshi et al., 2019) and the pykonal library (White et al., 2020) for FMM on $100 \times 100 \times 100$ resolution grids. All experiments were performed on a device with a 3.50GHz × 8 Intel Core i9 processor, 32 GB RAM, and GeForce RTX 3090 GPU. We use 1 million start and goal pairs to train our models in environments besides Gibson scenes, and we use 2 million start and goal pairs in Gibson scenes. Furthermore, on our system, the training took around 5-16 hours for each model.

| Env | Model | Data Generation Time | Training Time |
|---|---|---|---|
| Box | NTFields | 3.90s | 1500 epochs 4.9h |
| Bunny | NTFields | 14.15s | 1500 epochs 4.9h |
| Cluttered 3D | NTFields | 3.74s | 2000 epochs 7.2h |
|  | IEF3D | 2.6h | 2000 epochs 7.8h |
| Gibson 1 | NTFields | 87.49s | 2000 epochs 15.9h |
|  | IEF3D | 2.6h | 2000 epochs 16.7h |
| Gibson 2 | NTFields | 130.36s | 2000 epochs 15.9h |
|  | IEF3D | 2.6h | 2000 epochs 16.7h |
| 4-DOF Manipulator | NTFields | 54.86s | 1200 epochs 8.5h |
| 6-DOF Manipulator | NTFields | 41.33s | 1200 epochs 10.5h |

Table 1: Data Generation and Training Time

We run FMM on the grid with resolutions 11, 21, and 51 on 4, 5, and 6 DOF robot manipulators to show its computational time and corresponding approximate IEF3D data preparation times in Table 4.

| Resolution | 4-DOF | | 5-DOF | | 6-DOF | |
|---|---|---|---|---|---|---|
| 11 | 0.008s | 2.2h | 0.15s | 41.7h | 3.74s | 43.3d |
| 21 | 0.12s | 33.3h | 8.19s | 94.8d | 382.5s | 4427d |
| 51 | 10.74s | 124.3d | - | - | - | - |

Table 2: Resolution shows the number of divisions along each axis. The higher number, the better. Under each robot system, the left column shows the FMM running time from a source point, and the right column shows approximate data generation times for IEF3D. The notations $s$, $h$, and $d$ denote seconds, hours, and days, respectively. The empty parts in the table mean running out of memory.

### E.1    Cluttered 3D Experiments

We run our method on multiple cluttered 3D environments. The speed and time fields' visualization is shown in Fig. 9. In all eight environments, our method recovers similar fields as FMM, thus validating the correctness of our time-field learning approach. Table 5 shows the statistical results of our method and traditional planning techniques in these eight environments.

|          | time (sec)      | length            | safe margin     | sr(%) |
|----------|-----------------|-------------------|-----------------|-------|
| NTFields | $0.05 \pm 0.00$ | $15.48 \pm 15.75$ | $2.09 \pm 1.06$ | 99.95 |
| IEF3D    | $0.04 \pm 0.00$ | $15.48 \pm 15.75$ | $2.09 \pm 1.06$ | 96.35 |
| FMM      | $0.58 \pm 0.02$ | $15.45 \pm 15.59$ | $2.10 \pm 1.06$ | 100   |
| RRT*     | $1.51 \pm 0.00$ | $14.43 \pm 12.16$ | $0.61 \pm 2.45$ | 100   |
| LazyPRM* | $0.82 \pm 0.00$ | $14.10 \pm 11.48$ | $0.85 \pm 2.52$ | 100   |

Table 3: The table shows statistical results on 1000 different starts and goals for eight different scenes.

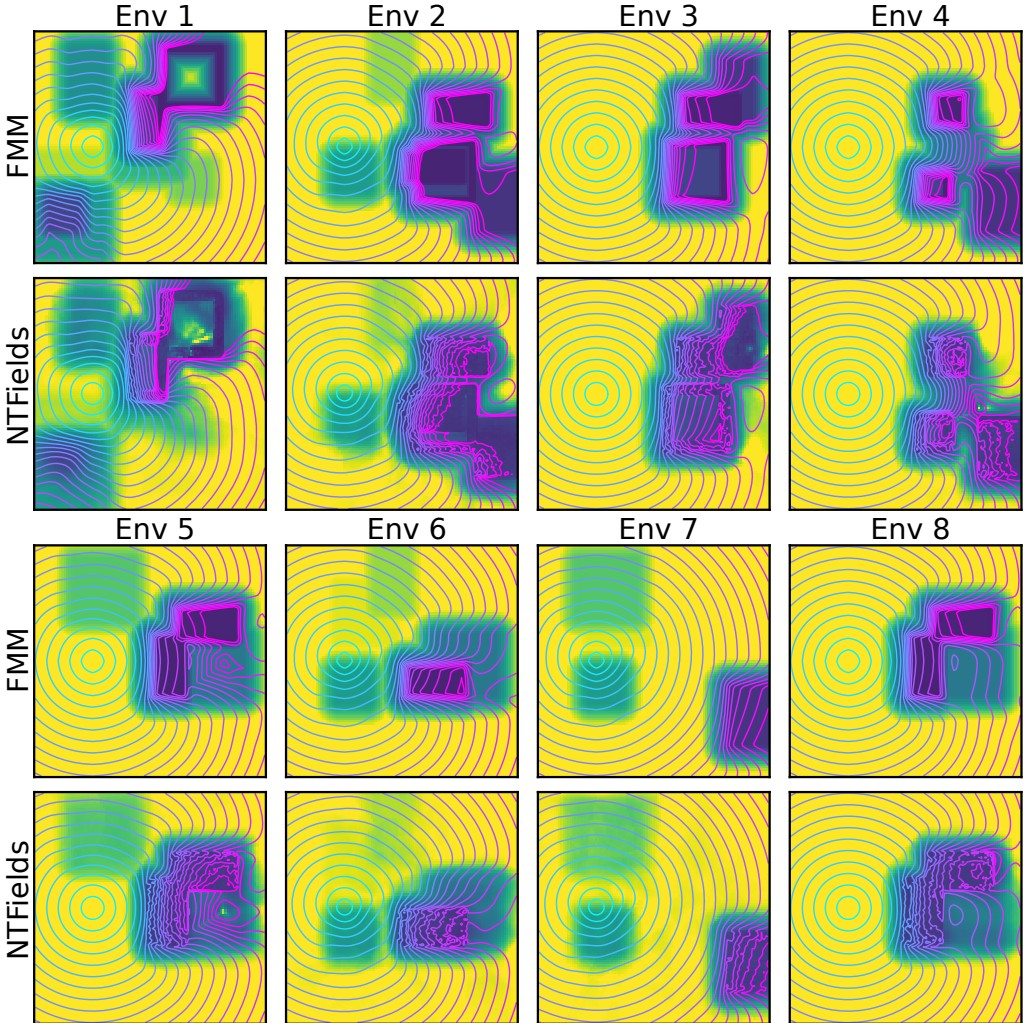

Figure 9: Speed Field and Time Field visualization, the colors show speed, and contour lines show arrival time from a source point, comparing NTFields (our) and FMM in eight different cluttered 3D environments

### E.2 GIBSON EXPERIMENTS

We use a Gibson scene (Fig. 10) to discuss the design of our method and our choice of w-space encoder training parameters $l_0$ and $l_1$. The value of $l_0$ is selected based on time-field results. We let the time-field net and c-space encoder train end-to-end without the w-space encoder until they learn to recover time fields to some extent. We observed that we usually get an approximate time field in about 500 epochs in all cases. Therefore $l_0$ is set to 500. The $l_1$ is set larger than $l_0$. In our

experiments, it is set to 1000. Thus, the w-space encoder begins training after $l_0$. Fig. 10 indicates that directly adding w-encoder (NTFields w/), $l_0$, and $l_1 = 0$, will lead to the local minimum. Without w-encoder ((NTFields w/o), i.e., $l_0$ =inf, it can lead to the correct result but takes longer to converge. Since we want to make our model converge faster, we add a w-encoder after $l_0 = 500$ (NTFields), which converges to recover the time field quickly and efficiently.

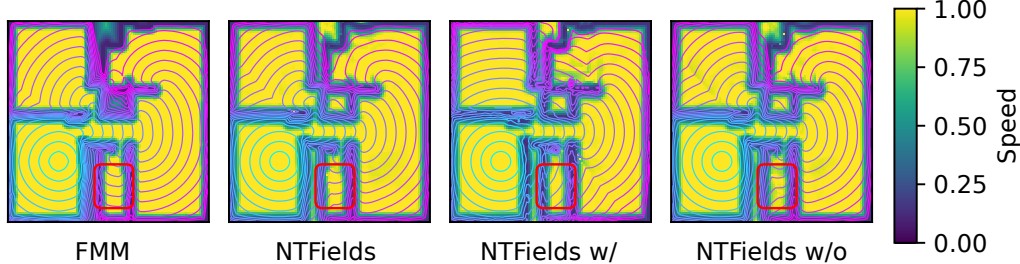

Figure 10: Speed Field and Time Field visualization. The colors show speed, and contour lines show arrival time from a source point, comparing NTFields, NTFields with w-encoder (NTFields w/), NTFields without w-encoder (NTFields w/o), and FMM in a Gibson environment

