# OpenReview forum: "NTFields: Neural Time Fields for Physics-Informed Robot Motion Planning"
_ICLR.cc/2023/Conference — ICLR 2023 notable top 25%_

### Official Review · Reviewer_t8K7 · 2022-10-23

**Confidence:** 4
**Correctness:** 3
**Technical Novelty And Significance:** 3
**Empirical Novelty And Significance:** 3
**Recommendation:** 8

**Clarity, Quality, Novelty And Reproducibility:**

Clarity: The paper is well-written
Quality: The technique proposed is sound
Originality: good.

**Strength And Weaknesses:**

Strength:

1. The paper is well-written and well-motivated.

2. The algorithm is novel to me. Incorporating physics prior to alleviate the costly computation is interesting.

3. The techniques are sound and performance on the benchmark tasks are good.

Weakness:

1. The deduction from Equ. 1 to Equ. 2 can be clearer or supplemented in appendix.

2. Will the choice of ``s_const’’ in speed model influence the training? I think change of s_const will also change the numeric distribution of  corresponding speed field and time field.

3. Adopting Eikonal equation as physics prior is actually a little arbitrary. I do not see the physics connection between a wave propogation model to a planning problem. Does the trajectory of wave propogation guarantee an optimal trajectory path? Apparently no. It is also confirmed in the result tables. It does not generate the shortest path with highest success rate.

4. Analysis on failure cases would be nice.

5. Typos:

    a) In many places, “expensive” is written as “expansive”

**Summary Of The Paper:**

This paper proposes a novel neural motion planning algorithm based on Eikonal equations. It requires no trajectory data generation in advance for training, and has great performance in both cluttered environments and high-dof manipulation cases.

**Summary Of The Review:**

I think this paper proposes a novel and interesting framework for motion planning with physics-informed framework.
Though I do think there are some places of the paper can be polished, the overall is sound.

---

> ### Author Response · Authors · 2022-11-09
> **For Reviewer t8K7**
>
> We would like to thank our reviewer for their feedback. We have revised the paper accordingly and all changes are highlighted in red in our revised paper.
> >The deduction from Equ. 1 to Equ. 2 can be clearer or supplemented in appendix.
>
> We have included the derivation of obtaining Eq2 from Eq1 in Appendix A
> >Will the choice of ``s_const’’ in speed model influence the training? I think change of s_const will also change the numeric distribution of corresponding speed field and time field.
>
> s_const will not affect results, because it will be scaled to one for training. However, d_min will affect the minimal speed on obstacles, and d_max will affect the safety zone near obstacles.
> >Adopting Eikonal equation as physics prior is actually a little arbitrary. I do not see the physics connection between a wave propogation model to a planning problem. Does the trajectory of wave propogation guarantee an optimal trajectory path? Apparently no. It is also confirmed in the result tables. It does not generate the shortest path with highest success rate.
>
> We revised the Eikonal equation part in our paper and show our motivation to use it. The Eikonal equation is a first-order, nonlinear PDE that approximates wave propagation. Its solution is the shortest arrival time from a source to a destination location under a pre-defined speed model. Furthermore, the Eikonal equation is known for solving the continuous shortest path planning problems [1,2] with applications to robot motion planning [3].
>
> The lower success rate is due to the imperfection of the neural network PDE solver. If it can get the exact solutions of the Eikonal equation, the result should be optimal [2]. Since FMM discretized space, it may lose some information about the environment. In contrast, the sampling-based methods sample in continuous space, which may lead to a higher success rate in some scenarios under probabilistic guarantees.
>
> >Analysis on failure cases would be nice.
>
> Since neural networks (NNs) are approximation models, there will always be some error in learning the time-field model using the Eikonal Equation. Therefore, in our setting, most failures occur when the NN fails to recover the time field in a certain part of the given environment. We leave further investigation into this aspect of NNs to our future work.
> >Typos:
> a) In many places, “expensive” is written as “expansive”
>
> fixed
>
> [1] Sethian, J. A. (1996). "A Fast Marching Level Set Method for Monotonically Advancing Fronts"
>
> [2] Clawson, Z.; Chacon, A.; Vladimirsky, A. (2014). "Causal Domain Restriction for Eikonal Equations". SIAM Journal on Scientific Computing
>
> [3] Alberto Valero-Gomez, Javier V Gomez, Santiago Garrido, and Luis Moreno. The path to efficiency: Fast marching method for safer, more efficient mobile robot trajectories. IEEE Robotics & Automation Magazine, 20(4):111–120, 2013.

---

> > ### Comment · Reviewer_t8K7 · 2022-12-11
> > **Rating after rebuttal.**
> >
> > I appreciate the author's rebuttal, which addresses most of my concerns. After carefully reading other reviewers' comments, I decided to raise my rating to accept.

---

> > > ### Author Response · Authors · 2022-12-12
> > > **Thanks for your response**
> > >
> > > Thank you very much for taking the time to reassess our work.

---

> ### Author Response · Authors · 2022-11-18
> **Looking forward to discussion and reassessment of our work**
>
> Dear reviewer, thank you for your time in reviewing our work and providing constructive feedback. We have made revisions and offered detailed clarification to current questions. If our response and the revised manuscript address your concerns, we would be grateful if you could reassess our work.

---

### Official Review · Reviewer_8txW · 2022-10-23

**Confidence:** 3
**Correctness:** 4
**Technical Novelty And Significance:** 3
**Empirical Novelty And Significance:** 2
**Recommendation:** 6

**Clarity, Quality, Novelty And Reproducibility:**

### Clarity & Reproducibility

- This paper is well written, and it is possible to reproduce this work using the detailed description.

### Quality

- The paper included plenty of information on a physics-informed neural motion planner. The equations and materials are appropriately placed and easy to follow. Solving the problems of robot motion planning can be a good contribution for ICLR.

### Novelty

- Currently, I am not completely certain about the novelty of this work since there might be other continous time motion planning models. I am looking forward to discuss this issue with authors and other reviewers in the discussion phase.

**Strength And Weaknesses:**

## Strengths

- As far as I understand, the latent embedding of the configuration-space, that is processed with a nerual network, is novel. I believe this work contibutes to general stuides of motion planning and representation learning.
- The overall methodology is sound, and there are some experiments that support the authors’ claim.

## Weaknesses

- Even though I think using a physics-insipred differential equation in a model is a good idea, but I think current version of manuscript the paper does not fully cover the reasoning behind introducing an eikonal equation in the motion planing domain. The authors are encouage to elaborate on the motivation of the equation for solving the navigation problem and fundamental benefits of using them over other planning methods for continous spaces.
- In the article, it is not clear whether introducing a space encoder f(q) enables a finer planning results theoretically; likewise, I have an impression that the encoding method for the C-space seems to be not carefully designed to solve this specific problem.
- In experiments, the performance of NTFileds (Figs. 4 & 6)  is not stellar. The authors are encouraged to put effort on validating their methodology qualitatively in a real-world problem.

**Summary Of The Paper:**

This paper proposed a novel method of neural motion planning called neural time fields (NTFields) for robot navigation tasks. The key idea is to use an eikonal equation for modeling a wave propgation of arrival time, which probably corresponds to the shortest path from the origin (and the target) to each arbitrary continuous points using a wave propgation. The model is composed of three subnetworks: (1) C-space encoder, (2) workspace encoder, and (2) time field generator. The C-space and workspace encoders encode the general goal configuration and robot configuration. Using these representations, the time field generator conducts biddirectional (both direction from the start and the target) motion planning and updates the configurations via propagation of each step, without expert data.

**Summary Of The Review:**

Before the authors’ response, I have noted some concerns regarding motivation, methodology, experiments, and novelty. I will refine this assessment based on the response and discussion.

---

> ### Author Response · Authors · 2022-11-09
> **For Reviewer 8txW**
>
> We would like to thank our reviewer for their feedback. We have revised the paper accordingly and all changes are highlighted in red in our revised paper.
> >Even though I think using a physics-insipred differential equation in a model is a good idea, but I think current version of manuscript the paper does not fully cover the reasoning behind introducing an eikonal equation in the motion planing domain. The authors are encouage to elaborate on the motivation of the equation for solving the navigation problem and fundamental benefits of using them over other planning methods for continous spaces.
>
> We revised the Eikonal equation part in our paper and show our motivation to use it. The Eikonal equation is a first-order, nonlinear PDE that approximates wave propagation. Its solution is the shortest arrival time from a source to a destination location under a pre-defined speed model. Furthermore, the Eikonal equation is known for solving the continuous shortest path planning problems [1,2] with applications to robot motion planning [3].
> >In the article, it is not clear whether introducing a space encoder f(q) enables a finer planning results theoretically; likewise, I have an impression that the encoding method for the C-space seems to be not carefully designed to solve this specific problem.
>
> We carefully designed f(q), and here is our justification. To train and use our time fields model, we need to compute the gradient of time with respect to input configurations. Therefore, we directly encode q with MLP-based ResNet blocks to allow gradient flow for gradient computation. To highlight this aspect, we have revised our description in the paper related to f(q).
> >In experiments, the performance of NTFileds (Figs. 4 & 6) is not stellar. The authors are encouraged to put effort on validating their methodology qualitatively in a real-world problem.
>
> We leave the extension of our method to real-robot systems to our future work.
> >Novelty: Currently, I am not completely certain about the novelty of this work since there might be other continous time motion planning models. I am looking forward to discuss this issue with authors and other reviewers in the discussion phase.
>
> To the best of our knowledge, our method is the first approach that uses deep neural networks to directly solve Eikonal Equation for robot motion planning. Other reviewers also agreed that our work is novel.
>
> [1] Sethian, J. A. (1996). "A Fast Marching Level Set Method for Monotonically Advancing Fronts"
>
> [2] Clawson, Z.; Chacon, A.; Vladimirsky, A. (2014). "Causal Domain Restriction for Eikonal Equations". SIAM Journal on Scientific Computing
>
> [3] Alberto Valero-Gomez, Javier V Gomez, Santiago Garrido, and Luis Moreno. The path to efficiency: Fast marching method for safer, more efficient mobile robot trajectories. IEEE Robotics & Automation Magazine, 20(4):111–120, 2013.

---

> > ### Comment · Reviewer_8txW · 2022-11-16
> > **Thanks for the clarifications**
> >
> > Dear Authors, thanks for the response and the revised manuscript. I will take these into account when discussing the paper with the other reviewers. I don't have any additional questions at this point.

---

> > > ### Author Response · Authors · 2022-11-16
> > > **Thanks for your response**
> > >
> > > We thank the reviewer for their response. Please let us know if anything needs further clarification.

---

### Official Review · Reviewer_9pD7 · 2022-10-24

**Confidence:** 4
**Correctness:** 4
**Technical Novelty And Significance:** 3
**Empirical Novelty And Significance:** 3
**Recommendation:** 8

**Clarity, Quality, Novelty And Reproducibility:**

The paper is generally well-written and novel, except for a few clarity issues:
- Sec 3.1: the term “arrival time-space” is not described.
- Sec 3.2: when talking about the encoding for robot configuration and workspace, the motivation for introducing them is not described anywhere before.

For reproducibility, it seems challenging to replicate the whole technical framework with appropriate details unless the authors are willing to open-source the code.

--------------------------------------
### Post-rebuttal update:
The authors promised to release the source code.

**Strength And Weaknesses:**

### Strengths:
- A novel technical approach formulation based on physics-informed learning for motion planning. This perspective is interesting.
- Empirically better field generation quality and faster planning time in execution given random start and goal configurations.

### Weaknesses:
- Experimental comparisons are weak. In my opinion, it can be much improved in several ways. See more details below.
- Expensive training time as shown in Table 3 of the Appendix. Though the data generation is fast, training takes on the magnitude of hours. In this case, given the budget of hours, FMM or IEF3D might not be computationally intractable, so more comparisons are needed.
- Not clear if the learned time field can generalize to different scenes (as IEF does). Do we need to retrain from scratch (for hours) when the underlying scene/layout changes?
- No explicit discussions on limitations and failure cases - this is very important for a fair understanding of the paper.

### Experimental comparison issues:
- It would be better to show qualitative comparisons on field generation on more examples (only 2 currently) or present quantitative metrics to support the claim that NTFields generate correct time fields.
- Lack of environment variations in the cluttered 3D environments - what would happen for different layouts? Also, the performance metrics “length” and “safe margin” seem somewhat contradicting to me, so not sure what the preference is here and if the proposed approach can control the trade-off.
- As mentioned above, FMM and IEF3D should be included in the comparison on the 4-DOF and 6-DOF manipulation examples, as the training time for NTFields takes 15+ hours. It’s necessary to see under the same generation+training time budget, how good FMM and IEF3D can achieve (the grid resolution does not need to be very high if they are indeed intractable, say it takes hundreds or more hours to generate the data).

### Questions:
- Does the bi-directional gradient-following approach always converge to a single path? Have you seen any cases where following the gradient will get stuck in a local plateau?
- Regarding the training details in Appendix D, how are the hyper-parameters of l_0 and l_1 chosen for different examples, or how to adjust them in a generalizable way?
- The nonlinear symmetric operator seems to help a little when looking at Figure 3. However, it is not clear how necessary it is for more examples and if it introduces extra training overheads.

--------------------------------------
### Post-rebuttal update:
The authors addressed most of my concerns described above, except for some major limitations of the method.

**Summary Of The Paper:**

This paper proposes Neural Time Fields (NTFields), A novel physics-informed formulation of learning the continuous arrival time field for robot motion planning. NTFields generate accurate time fields and plans faster than existing baselines in cluttered 3D environments.

**Summary Of The Review:**

This paper proposes a novel perspective and technical framework for learning physics-informed neural time fields for motion planning. However, I believe there is much room for improvement in the experimental comparison, and it lacks enough discussions on limitations.

--------------------------------------
### Post-rebuttal update:
The authors improved the experimental comparison and included discussions on limitations. Given the novelty, the sound technical approach, and adequate experimental comparison, I recommend acceptance. In my opinion, the major limitations are computational overhead and generalization to different scenes, which deserve to be future work.

---

> ### Author Response · Authors · 2022-11-09
> **For Reviewer 9pD7 (Part 2)**
>
> >As mentioned above, FMM and IEF3D should be included in the comparison on the 4-DOF and 6-DOF manipulation examples, as the training time for NTFields takes 15+ hours. It’s necessary to see under the same generation+training time budget, how good FMM and IEF3D can achieve (the grid resolution does not need to be very high if they are indeed intractable, say it takes hundreds or more hours to generate the data).
>
> Regarding FMM, it is not computationally tractable in high DOF settings. We report FMM results in 4, 5, and 6 DOF with three different resolution sizes in Appendix E, Table 4. In 6 DOF, the lower resolution takes around 4 seconds to compute a path, for the higher resolution, our system runs out of memory. Even if we consider the 6DOF case lower resolution case, IEF3D would take 1 million demonstrations to train, which corresponds to 1100 computational hours. Please also note that the training time of our method and IEF3D is similar due to the similar number of neural network parameters. Furthermore, for each new start configuration, we will have to run FMM from scratch, and our method, once trained, can generalize to the new start and goal pairs with little planning time. Finally, since running FMM is time-consuming, gathering data for IEF3D for manipulators is not computationally feasible.
> >Does the bi-directional gradient-following approach always converge to a single path?
>
> Yes, bidirectional means if we exchange the start and goal points, the path will be the same.
> >Have you seen any cases where following the gradient will get stuck in a local plateau?
>
> No, we haven’t observed cases where the gradient will stuck in a local plateau during planning. The failure is primarily due to poor speed fields in certain parts of environments. Furthermore, our symmetric operator also aids in bidirectional planning, i.e., if we swap start and goal, the arrival time needs to be the same.
> >Regarding the training details in Appendix D, how are the hyper-parameters of l_0 and l_1 chosen for different examples, or how to adjust them in a generalizable way?
>
> The value of l_0 is selected based on time-field results. We let the time-field net and c-space encoder train end-to-end without the w-space encoder until they learn to recover time fields to some extent. We observed that we usually get an approximate time field in about 500 epochs in all cases. Therefore l_0 is set to 500. The l_1 is set larger than l_0. In our experiments, it is set to 1000. Thus, the w-space encoder begins training after l_0. In Appendix E.2, we also show additional experiments indicating that directly adding w-encoder, l_0, and l_1 =0, will lead to the local minimum. Without w-encoder, i.e., l_0 =inf, it can lead to the correct result but takes longer to converge. Since we want to make our model converge faster, we add a w-encoder after l_0=500, which converges to recover the time field quickly and efficiently.
> >The nonlinear symmetric operator seems to help a little when looking at Figure 3. However, it is not clear how necessary it is for more examples and if it introduces extra training overheads.
>
> It does not add an extra training load. Furthermore, it aids in bidirectional planning, as described above. Also, we believe it is crucial to use PDE properties when designing a neural network architecture.
> >Sec 3.1: the term “arrival time-space” is not described.
>
> We have removed the term arrival-time space. Instead, we formally introduce the term arrival time as follows: The Eikonal equation is a first-order, nonlinear PDE that approximates wave propagation. Its solution is the shortest arrival time corresponding to the shortest path [3, 4] from a source to a destination location under a pre-defined speed model. (We have revised section 3.1)
> >Sec 3.2: when talking about the encoding for robot configuration and workspace, the motivation for introducing them is not described anywhere before.
>
> We have included the motivation, see Section 3.2. The former enables reasoning about the relative position of obstacles and the robot, including collisions, for decision-making. Whereas the latter allows direct computation of the gradient of time concerning input configurations needed for solving the Eikonal Equation.
>
> >For reproducibility, it seems challenging to replicate the whole technical framework with appropriate details unless the authors are willing to open-source the code.
>
> We will open-source our implementation.
>
> [1] James Bradbury, Roy Frostig, Peter Hawkins, Matthew James Johnson, Chris Leary, Dougal Maclaurin, and Skye Wanderman-Milne. JAX: composable transformations of Python+NumPy programs, 2018.
>
> [2] Sethian, J. A. (1996). "A Fast Marching Level Set Method for Monotonically Advancing Fronts"
>
> [3] Clawson, Z.; Chacon, A.; Vladimirsky, A. (2014). "Causal Domain Restriction for Eikonal Equations". SIAM Journal on Scientific Computing

---

> > ### Comment · Reviewer_9pD7 · 2022-11-14
> > **Thanks for your response**
> >
> > Thank you for incorporating the suggestions and making thoughtful revisions. Overall, most of my concerns are addressed, except for a few described below.
> >
> > It’s good to see that the authors provided more qualitative comparisons of generated time fields vs ground truth in more environments, especially cluttered 3D envs. But as requested, I would like to see more statistical comparisons like the table shown in Fig 4 between different methods to strengthen the experimental comparison, if time permits.
> >
> > I appreciate the authors elaborating more details on the slow data generation speed of FMM in high-dim environments. As a side comment, apart from the proposed approach, is there any other relevant method that uses deep learning to solve the Eikonal equation approximately? In that case, generating time fields (approximately) might be tractable in high-dimensional problems.
> >
> > (Will update the score as soon as I am allowed to do that)

---

> > > ### Author Response · Authors · 2022-11-16
> > > **Thanks for your response**
> > >
> > > We are highly thankful to the reviewer for their prompt response and feedback. We add a new table in Appendix E comparing our method, FMM, RRT*, and LazyPRM*. Unfortunately, we won't be able to finish IEF3D and MPNet data generation, training, and testing by this week. However, it can be seen that our method retains the computational benefits over the traditional methods by a significant margin across different environments.
> > >
> > > For the Eikonal equation, as far as we know, previous methods focus on the sound wave in geophysics [1] or geodesic line on surface mesh [2], both of which are 3D problems. And, to the best of our knowledge, we are the first to solve Eikonal Equation using neural networks in the high dimensional problems corresponding to robotics applications.
> > >
> > > [1] Jonathan D Smith, Kamyar Azizzadenesheli, and Zachary E Ross. EikoNet: Solving the eikonal equation with deep neural networks. IEEE Transactions on Geoscience and Remote Sensing.
> > >
> > > [2] M. Lichtenstein, G. Pai, and R. Kimmel, “Deep Eikonal solvers,” in Scale Space and Variational Methods in Computer Vision. Cham, Switzerland: Springer, Jun. 2019, pp. 38–50.

---

> ### Author Response · Authors · 2022-11-09
> **For Reviewer 9pD7 (Part 1)**
>
> We would like to thank our reviewer for their feedback. We have revised the paper accordingly and all changes are highlighted in red in our revised paper.
>
> >Expensive training time as shown in Table 3 of the Appendix. Though the data generation is fast, training takes on the magnitude of hours. In this case, given the budget of hours, FMM or IEF3D might not be computationally intractable, so more comparisons are needed.
>
> During the paper submission time, our code could have been more optimized. We have revised the training times with our improved implementation, which we will also release with our accepted paper. Furthermore, since it’s a deep learning paper, it should be expected to have training times. IEF3D training times are also similar to our method since the number of neural network parameters are similar and in addition, it also requires demonstration data (see Table 3, Appendix E). We agree with the reviewer that training times need to be faster, but it’s beyond the scope of this paper. We should highlight that with advanced libraries like JAX [1], the training times can further be reduced for methods requiring gradient computation. We propose to explore such libraries in our future work. Regarding FMM, it is not computationally tractable in high DOF settings in our experiment. Please note that FMM requires space discretization, and for each new start configuration, we will have to run FMM from scratch, whereas our method, once trained, can generalize to the new start and goal pairs and finds paths very quickly. Appendix E, Tables 4 and 5 provide computational times for FMM with different resolution scales and estimated data generation time for IEF3D. It can be seen that IEF3D training times are similar to ours, but data generation times are extremely large.
>
>
> >Not clear if the learned time field can generalize to different scenes (as IEF does). Do we need to retrain from scratch (for hours) when the underlying scene/layout changes?
>
> Currently, our method only generalizes to new, unseen start and goal pairs in a given environment. We already highlight it in our conclusion and future work section (Sec 5).
> >No explicit discussions on limitations and failure cases - this is very important for a fair understanding of the paper.
>
> We have revised Section 5 to highlight limitations. First, currently, our method considers only the collision and speed constraints. Second, our method only generalizes to the new start and goal configurations in a given environment and we cannot generalize to different environments. Finally, since neural networks (NNs) are an approximation to PDE, there are few failure cases due to errors in learning the time-field model using the Eikonal Equation. In addition, we leave the extension of our method to real-robot systems to our future work.
>
> >It would be better to show qualitative comparisons on field generation on more examples (only 2 currently) or present quantitative metrics to support the claim that NTFields generate correct time fields. Lack of environment variations in the cluttered 3D environments - what would happen for different layouts?
>
> We have included various cluttered 3D layouts in Appendix E.1.
>
> >Also, the performance metrics “length” and “safe margin” seem somewhat contradicting to me, so not sure what the preference is here and if the proposed approach can control the trade-off.
>
> Note that the safe margin is due to our predefined speed model, which constraints speed around obstacles. We show that our paths are short under those constraints, leading to a smooth transition around obstacles. On the other hand, traditional methods like RRT* and PRM* assume constant speed in obstacle-free space and zero speed in obstacle space. Therefore, they lead to paths with sharp turns. However, with those methods, if we use our speed model, they can recover similar paths as ours with a safe margin, even though those traditional methods do not explicitly consider safety constraints.

---

### Official Review · Reviewer_E33u · 2022-10-24

**Confidence:** 4
**Correctness:** 3
**Technical Novelty And Significance:** 3
**Empirical Novelty And Significance:** 3
**Recommendation:** 8

**Clarity, Quality, Novelty And Reproducibility:**

The paper is very well-written and easy to follow. The paper discusses several contributions in the context of the introduced NTFields framework and the results are promising. Enough details are provided to reproduce the discussed method.

Specific comments:

- In Section 3.1. it is not clear what is meant by 'arrival time-space'. Please clarify, or introduce more formally, also w.r.t. the wave propagation.
- Section 3.1. 'are not trivial' to obtain?
- 'First, we sample the sparse workspace obstacle point cloud ... and covert them' --> and convert it
- Figure 1: It would be helpful to more clearly visualize q_s and q_t in the corresponding maps. It seems q_s is blue and q_t is red, but it would be helpful to add labels to the speed, time and bidirectional maps.
- The sample generation on the robot surface is not clear. Why is it necessary to do this with forward kinematics? Please provide more details.
- C-space encoding: 'It takes the robot configuration q as input'. Network f is a ResNet, so q needs to be a map (image)? Please also show that in Figure 1 or as an additional figure.
- Section 4.2 (Large Complex Environments), P2: 'The paths shown in 6' --> Fig. 6
- It is not clear what is shown in Figure 4. Are the gray blocks obstacles or the environment? Please add a clarifying statement to the caption.
- Figure 7: It would be helpful if the caption of this figure would more carefully explain what is shown.

**Details Of Ethics Concerns:**

Motion planing for autonomous agents is established research direction in the field of robotics. I do not have concerns toward ethical considerations.

**Strength And Weaknesses:**

Strengths:

- The paper addresses an important problem with a sound technical solution.
- The paper is very well written and easy to read.
- The paper discusses a set of meaningful experiments that show the benefits of the introduced NMP based on the NTField framework.

Weaknesses:

- The paper does not discuss any limitations.
- The method is only validated on synthetic data and not on real robots.

**Summary Of The Paper:**

The proposed paper introduces a novel method for robot motion planing based on a wave propagation model. The introduced model generates path solutions based on a non-linear first-order PDE (Eikonal Equation). The method is validated for motion planning tasks in cluttered 3D environments (Gibson). The discussed results indicate high success rates and lower computation times compared to SOTA methods.

**Summary Of The Review:**

Overall, this is an interesting paper that can be accepted to ICLR 2023. The paper is very well written, the experiments are sound, and the problem the proposed method addressees is important. The proposed NTField framework is a novel contributions that also has the potential to stimulate future work in the field of motion planing for robotics. Furthermore, it seems that physics-informed approaches are an important and novel direction in the field of robotics. An interesting direction (for future work) would be to employ the proposed framework for real world robotics experiments. Finally, I would like to encourage the authors to carefully discuss the limitations of the propose method.

---

> ### Author Response · Authors · 2022-11-09
> **For Reviewer E33u**
>
> We would like to thank our reviewer for their feedback. We have revised the paper accordingly and all changes are highlighted in red in our revised paper.
>
> >Weaknesses:
> The paper does not discuss any limitations. The method is only validated on synthetic data and not on real robots.
>
> We have revised Section 5 to highlight limitations. First, currently, our method considers only the collision and speed constraints. Second, our method only generalizes to the new start and goal configurations in a given environment and we cannot generalize to different environments. Finally, since neural networks (NNs) are an approximation to PDE, there are few failure cases due to errors in learning the time-field model using the Eikonal Equation. In addition, we leave the extension of our method to real-robot systems to our future work.
>
> >In Section 3.1. it is not clear what is meant by 'arrival time-space'. Please clarify, or introduce more formally, also w.r.t. the wave propagation.
>
> We have removed the term arrival-time space. Instead, we formally introduce the term arrival time as follows: The Eikonal equation is a first-order, nonlinear PDE that approximates wave propagation. Its solution is the shortest arrival time corresponding to the shortest path [1, 2] from a source to a destination location under a pre-defined speed model. [We have revised section 3.1 to describe the notion of arrival-time]
> >Section 3.1. 'are not trivial' to obtain?
>
> We meant it's computationally expansive to obtain a time field, especially in high-dimensional problems. So, we have changed "not trivial' to "computationally expansive."
> >'First, we sample the sparse workspace obstacle point cloud ... and covert them' --> and convert it
>
> Fixed
> >Figure 1: It would be helpful to more clearly visualize q_s and q_t in the corresponding maps. It seems q_s is blue and q_t is red, but it would be helpful to add labels to the speed, time and bidirectional maps.
>
> We have revised Fig. 1 accordingly
> >The sample generation on the robot surface is not clear. Why is it necessary to do this with forward kinematics? Please provide more details.
>
> FK allows computing robot joint positions in the workspace using the robot joint angles. Given all joints' positions and robot geometry, i.e., links' lengths and widths, we generate samples on the robot surface. Note that the robot surface points are in the workspace, as is the obstacle point cloud. Therefore, using both robot surface point cloud and obstacle point cloud to obtain workspace encoding enables reasoning about the relative position of obstacles and the robot itself, including collisions, for decision-making.  We have revised our description of w-space encoding in Sec 3.2.
>
> >C-space encoding: 'It takes the robot configuration q as input'. Network f is a ResNet, so q needs to be a map (image)? Please also show that in Figure 1 or as an additional figure.
>
> By ResNet-style, we mainly meant that our network has ResNet-style blocks with skip connections, but instead of CNN, we use MLP, f(q), to encode D dimensional robot configuration vector q.  We have revised our description in the paper accordingly (see section 3.2 ).
> >Section 4.2 (Large Complex Environments), P2: 'The paths shown in 6' --> Fig. 6
> Fixed
> >It is not clear what is shown in Figure 4. Are the gray blocks obstacles or the environment? Please add a clarifying statement to the caption.
>
> The gray blocks are the obstacles. We make them slightly transparent for visualization purposes only. We have revised our caption accordingly.
> >Figure 7: It would be helpful if the caption of this figure would more carefully explain what is shown.
>
> We have revised the caption
>
> [1] Sethian, J. A. (1996). "A Fast Marching Level Set Method for Monotonically Advancing Fronts"
>
> [2] Clawson, Z.; Chacon, A.; Vladimirsky, A. (2014). "Causal Domain Restriction for Eikonal Equations". SIAM Journal on Scientific Computing

---

> > ### Comment · Reviewer_E33u · 2022-12-01
> > **Response**
> >
> > Thank you for providing the additional information and for adding the clarifications to the paper. I do not have further questions.

---

> ### Author Response · Authors · 2022-11-18
> **Looking forward to discussion and reassessment of our work**
>
> Dear reviewer, thank you for your time in reviewing our work and providing constructive feedback. We have made revisions and offered detailed clarification to current questions. If our response and the revised manuscript address your concerns, we would be grateful if you could reassess our work.

---

> > ### Comment · Reviewer_E33u · 2022-12-01
> > **Response**
> >
> > Thank you for improving your working according to the comments in the reviews. I am now in full support of accepting this work and changed my score accordingly.

---

> > > ### Author Response · Authors · 2022-12-02
> > > **Thanks for your response**
> > >
> > > Thank you very much for taking the time to reassess our work.

---

### Author Response · Authors · 2022-11-09
**Summary for all reviewers and area chair**

We want to thank all reviewers for their detailed and constructive feedback. We provide individual responses to each reviewer's comments. The corresponding changes in the paper are highlighted in red for reviewers' convenience. In addition to addressing all minor concerns and clarification questions, we also highlight the following:

- We show that IEF3D and our method have similar training times due to the equivalent number of neural network parameters. However, the data generation time for our approach is at max 3 minutes. In contrast, for IEF3D, we show with experiments based on FMM that it can take up to several days in higher dimensional problems (See Appendix E).

- We highlight the limitations of our work and motivate our future work around them in Section 5.

- We also show that our training strategy of incorporating workspace and c-space encoders allows gradient computation and faster convergence to approximate the time fields for motion planning.

- We confirm that we will publicly release our source code implementing NTFields and related baselines in the given test cases with the final version of our manuscript.

---

### Author Response · Authors · 2022-11-14
**A kind reminder for author-reviewer discussion**

Dear reviewers, once again, thank you very much for your time and detailed feedback! Since the revision period will end this week, we would appreciate it if we could hear your response to our revised paper. We will be happy to address any further concerns and can modify our paper accordingly.

---

### Decision · Program_Chairs · 2023-01-20

**Decision:**

Accept: notable-top-25%

**Justification For Why Not Higher Score:**

It could contain more experiments on various environments and also different scenes. It would have been nice to see the discussion on its limitations too.

**Justification For Why Not Lower Score:**

The technical contribution is clear and it can lead to many follow-ups -- I believe this project deserves attention for others to think about.

**Metareview: Summary, Strengths And Weaknesses:**

This paper has positive reviews from all reviewers. It is clearly well written, and the proposed idea is novel as well. I debated whether this paper should be an oral or a spotlight; but because the draft as is is missing some deeper analysis (on various environments, for example), I think it is better to be a spotlight, despite its clear technical contribution.

**Note From Pc:**

if the above contains the word "oral" or "spotlight" please see: "oral" presentation means -> notable-top-5% and "spotlight" means -> notable-top-25%. As stated in our emails, we are disassociating presentation type from AC recommendations